# Retrospective and projected warming-equivalent emissions from global livestock and cattle calculated with an alternative climate metric denoted GWP*

**Agustin del Prado** [1,2]*, **Brian Lindsay**[3], **Juan Tricarico**[4]

1 Basque Centre for Climate Change (BC3), Edificio Sede no. 1, Planta 1, Parque Científico de UPV/EHU, Barrio Leioa, Bizkaia, Spain, 2 Ikerbasque—Basque Foundation of Science, Bilbao, Spain, 3 Global Dairy Platform, Rosemont, IL, United States of America, 4 Innovation Center for U.S. Dairy, Rosemont, IL, United States of America

* agustin.delprado@bc3research.org

**Data Availability Statement:** All the data and formulae used have been included in a spreadsheet as Supplem. material (SF1).

**Funding:** The authors of this paper report the following sources of funding: Global Dairy Platform

## Abstract

Limiting warming by the end of the century to 1.5˚C compared to pre-Industrial times requires reaching and sustaining net zero global carbon dioxide ($CO_2$) emissions and declining radiative forcing from non-$CO_2$ greenhouse gas (GHG) sources such as methane ($CH_4$). This implies eliminating $CO_2$ emissions or balancing them with removals while mitigating $CH_4$ emissions to reduce their radiative forcing over time. The global cattle sector (including Buffalo) mainly emits $CH_4$ and $N_2O$ and will benefit from understanding the extent and speed of $CH_4$ reductions necessary to align its mitigation ambitions with global temperature goals. This study explores the utility of an alternative usage of global warming potentials (GWP*) in combination with the Transient Climate Response to cumulative carbon Emissions (TCRE) to compare retrospective and projected climate impacts of global livestock emission pathways with other sectors (e.g. fossil fuel and land use change). To illustrate this, we estimated the amount and fraction of total warming attributable to direct $CH_4$ livestock emissions from 1750 to 2019 using existing emissions datasets and projected their contributions to future warming under three historical and three future emission scenarios. These historical and projected estimates were transformed into cumulative $CO_2$ equivalent ($GWP_{100}$) and warming equivalent (GWP*) emissions that were multiplied by a TCRE coefficient to express induced warming as globally averaged surface temperature change. In general, temperature change estimates from this study are comparable to those obtained from other climate models. Sustained annual reductions in $CH_4$ emissions of 0.32% by the global cattle sector would stabilize their future effect on global temperature while greater reductions would reverse historical past contributions to global warming by the sector in a similar fashion to increasing C sinks. The extent and speed with which $CH_4$ mitigation interventions are introduced by the sector will determine the peak temperature achieved in the path to net-zero GHG.

supported authors AdP and BL. Ikerbasque, Basque Foundation for Science supported AdP, Spanish National Plan for Scientific and Technical Research and Innovation supported AdP through grant (RYC-2017-22143), Ministerio de Ciencia e Innovación supported AdP through grant (CEX2021-001201-M), Eusko Jaurlaritza supported AdP through grant (BERC 2022-2024), Dairy Management Inc (US) supported AdP and JT through Global Dairy Platform AdP was also supported through Global Dairy Platform by Arla Foods, Dairy Australia, Dairy Companies of New Zealand, Global Round Table for Sustainable Beef, Innovation Centre for US Dairy, McDonalds Corporation, and Meat and Livestock Australia. BL is supported by Global Dairy Platform. JT received salary from Dairy Management Inc. The funders had a role in the study design by providing some of the general questions. The specific roles of these authors are articulated in the 'author contributions' section.

**Competing interests:** The authors have read the journal's policy and have the following competing interests: JT is a paid employee of Dairy Management Inc. There are no patents, products in development or marketed products associated with this research to declare.

## Introduction

The Intergovernmental Panel on Climate Change [1] estimated that the average temperature of the Earth's atmosphere and oceans increased by approximately 1.1˚C from 1850 to 2020. As part of the global effort to avoid dangerous consequences of climate change, the Paris Climate Agreement sets out a global framework to limit global warming to well below 2˚C and pursue efforts to limit it to 1.5˚C compared with preindustrial levels by 2050 [2]. In order to avoid surpassing these temperature limits, the article 4 of the Paris Agreement sets a greenhouse gas (GHG) mitigation goal of achieving 'a balance between anthropogenic emissions by sources and removals by sinks of GHGs'. One of the interpretations is that this balance can be reached with net-zero $CO_2$-equivalent emissions [3].

'Net-zero', is defined in the Working Group I contribution to the Sixth Assessment Report of the IPCC [1] for both carbon dioxide ($CO_2$) emissions (denoted as net-zero $CO_2$ or C neutrality) and for all GHG species (denoted as net-zero GHG or GHG neutrality). Net-zero $CO_2$ (in brackets for GHG) is the condition in which (metric-weighted) anthropogenic $CO_2$ (GHG) emissions associated with a reporting entity are balanced by (metric-weighted) anthropogenic $CO_2$ (GHG) removals over a specified period [1]. Temperature outcomes relative to net-zero GHG emissions will vary with the metric chosen to compare emissions and removals of different GHG [1].

Emission metrics provide a means of comparing different GHG emissions and removals by placing them on the same scale. This is typically done by quantifying a specified climate impact of a non-$CO_2$ gas relative to that of a $CO_2$ emission and is reported as '$CO_2$-equivalents' [4]. The most common GHG emission metric is the 100-year global warming potential ($GWP_{100}$) that compares the radiative forcing accumulated over a user-defined time-horizon (100 years) resulting from a pulse-emission of a specific GHG to a pulse-emission of an equal mass of $CO_2$ [4]. The development of alternatives referred to as 'step-pulse' emission metrics, such as CGTP [5] or GWP* [6], arises from the need to account for differences between the effects of long and short-lived GHGs on global temperature change [7]. Whereas methane ($CH_4$)'s impacts on temperature varies strongly with time after emissions occur due to its short atmospheric life, $CO_2$'s impact on temperature remains relatively constant for hundreds of years after the emission occurs [7]. Also, $CO_2$, once emitted, leads to increasing global temperature until net-zero $CO_2$ emissions are reached. By contrast, reductions in $CH_4$ emissions lead to reversing warming within a few decades. These differences in temperature change are hidden when calculating and using annual $CO_2$-e emissions to describe the impact of mitigation and targets based on aggregated annual emission rates [7]. In fact, GWP* was not developed to capture this behaviour, which was already well known, but to demonstrate that it could be quantified relatively simply while continuing to use the already familiar GWP concept [7].

It is recognized that $GWP_{100}$ shows the relative climate effect at one point in time resulting from an emission without needing comparison with past emissions [4] and is therefore inaccurate when estimating warming associated with emission time-series or pathways. This was highlighted by IPCC [1], which noted how $GWP_{100}$ either overestimates or underestimates global surface temperature changes depending on whether the $CH_4$ emissions rate was constant (overstated by a factor of 3–4) or increasing at high rates (understated by a factor of 4–5) over a 20-year time horizon [8]. Step-pulse metrics like GWP* can directly illustrate the anticipated temperature changes resulting from different emission pathways and incorporate them in 'cumulative emission budgets' or 'C budgets' to estimate how much each non-$CO_2$ gas contributes to warming [9] or remaining C budgets [10].

The agriculture sector is a large contributor of $CH_4$ emissions through rice cultivation and enteric fermentation and manure management from livestock. Livestock and global cattle (as defined by dairy and beef cattle and buffalo) contribute 30% and 24% of the global

anthropogenic $CH_4$ emissions, respectively [11]. Therefore, the choice of metric will strongly affect the calculated impact from $CH_4$ mitigation by the livestock and global cattle sectors as well as their contributions to historical warming from past emissions compared to $CO_2$ emission sources. These considerations could have substantial implications for climate goals, practice and policies recommendations, and their evaluation.

This article explores the impacts of acknowledging the differences between $CH_4$ and $CO_2$ using warming-equivalent emissions as calculated by GWP*. In concrete terms, we aimed to: (i) illustrate the use of GWP* instead of $GWP_{100}$ to compare the impact of $CH_4$ and $CO_2$ emission pathways on atmospheric warming using emissions from global fossil fuel $CO_2$, land use change $CO_2$ and livestock $CH_4$ reported in the scientific literature, (ii) investigate, using these metrics, the retrospective warming added by $CH_4$ emissions from livestock in aggregate and from global cattle (as defined by dairy and beef cattle and buffalo) globally between 1750–2019 and 1961–2019, respectively, and (iii) analyse, using these metrics, the implications on future global temperature change of adopting different $CH_4$ mitigation strategies that lead to warming stabilisation by the global cattle industry.

## Materials and methods

### Sources of emissions used for the retrospective analysis

The study uses several existing sources of $CH_4$ emissions data to illustrate the use of the GWP* climate metric to calculate warming-equivalent emissions and global temperature change from historical livestock and global cattle numbers. It must be noted that we used total livestock (including all animal species) $CH_4$ emissions for the long-term retrospective analysis (1750–2019) since data disaggregating livestock emissions by species (i.e. cattle) were not available for the years before 1961. For the short-term analysis (1961–2019) we used historical cattle (dairy and beef) and buffalo $CH_4$ emissions (all together referred as 'global cattle').

Long-term (1750–2019) historical livestock annual enteric and manure $CH_4$ emissions data were obtained from:

i. Reisinger and Clark [12] (1750–2009)

ii. EDGAR database https://edgar.jrc.ec.europa.eu/ (accessed on March 20, 2022) (2010–2019)

Short-term (1961–2019) historical cattle and buffalo annual enteric and manure $CH_4$ emissions were obtained from the FAOSTAT database [11]. For simplicity purposes, 'global cattle' refers to cattle (dairy and beef) and buffalo in this study.

A dataset comprising long-term (1750–2019) historical annual $CO_2$ emissions from fossil fuel (excluding carbonation) and land use change (LUC) from [13] was used for long-term (since 1750) historical comparisons with livestock $CH_4$ emissions and short-term (e.g. since 1961) historical comparisons with global cattle $CH_4$ emissions.

As mentioned above, long-term historical emissions data (including livestock) were considered since 1750 mainly because Friedlingstein et al. [13] includes data from this year which coincides with the beginning of the industrial revolution. In contrast, Reisinger and Clark [12] assumed no livestock emissions prior to 1860 in their study for practical reasons. Different alternative assumptions for livestock $CH_4$ emissions rates were simulated (details are explained in the scenario testing section below) to estimate the potential effects on global temperatures of including or excluding livestock emissions during the 1750–1859 period.

### Description of the data used for the retrospective analysis

The historical dataset used in our study shows that fossil $CO_2$ emissions increased steadily, especially since the mid-twentieth century, to approximately 36.7 Gt $CO_2$/yr by 2019 (Fig 1).

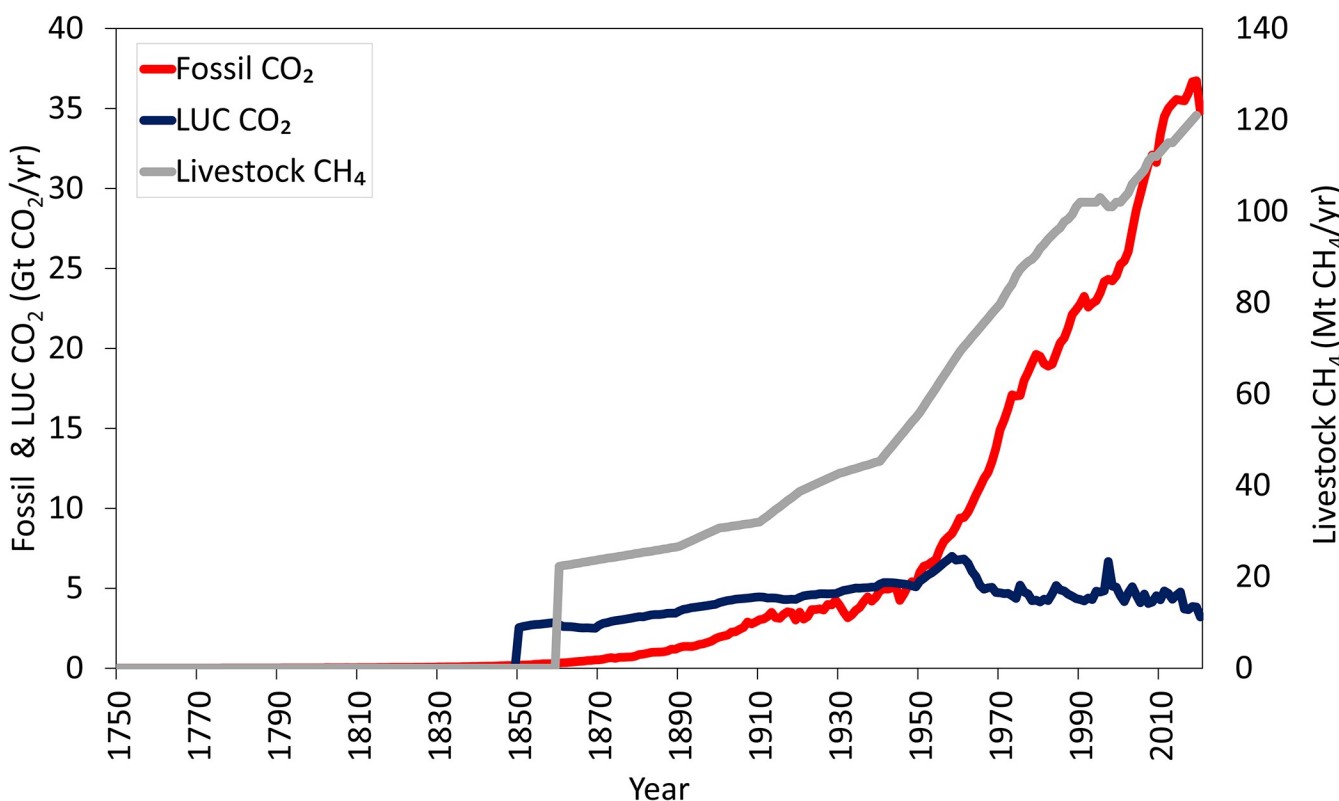

**Fig 1. Historical annual global emissions from fossil $CO_2$ use excluding carbonation (Fossil $CO_2$) and from land use change (LUC $CO_2$) (Gt $CO_2$ per year) and global $CH_4$ emissions from livestock (Mt $CH_4$ per year).** Note that $CO_2$ and $CH_4$ results are expressed in different units and each refers to the left and right y axes, respectively. Data source: $CO_2$: Friedlingstein et al. (2021) and $CH_4$: Reisinger and Clark (2018) (1750–2009) and EDGAR database (2010–2019).

In contrast, $CO_2$ emissions from LUC peaked in 1958 (7 Gt $CO_2$/yr) and 1997 (6.7 Gt $CO_2$/yr) but dropped during the last two decades to approximately 3.2 Gt $CO_2$/yr by 2019 (Fig 1). The dataset used in our study shows that long-term historical livestock $CH_4$ emissions increased from 22 to 121 Mt $CH_4$/yr from 1860 to 2019 (Fig 1).

The dataset used for short-term (1961–2019) historical global cattle $CH_4$ emissions shows they increased from approximately 59 Mt $CH_4$/yr in 1961 to almost 89 Mt $CH_4$/yr by 2019 (Fig 2A). In this dataset, if we disaggregate the data by animal types and sources, beef cattle represent the largest share of $CH_4$ emissions from the global cattle sector over 1961 to 2019 (Fig 2C), followed by dairy cattle (Fig 2B) and buffalo (Fig 2D). Enteric emissions represent approximately 94% of total $CH_4$ output by the global cattle sector with the remainder coming from manure management. The enteric contribution to total global cattle $CH_4$ emissions has grown from approximately 92% in 1961 to 95% in the last decade. The largest increases in $CH_4$ emissions occurred in the 1960s and 70s followed by stable emissions in the 90s and a more modest increase in the 2000s.

## $CO_2$-e and $CO_2$-we associated to using $GWP_{100}$ or GWP* to estimate global temperature changes

Cumulative $CO_2$ emissions show a near-linear relationship with their induced global warming [14]. This proportionality is represented by a coefficient that is referred to as TCRE (Transient Climate Response to cumulative carbon Emissions). This TCRE value can be multiplied by

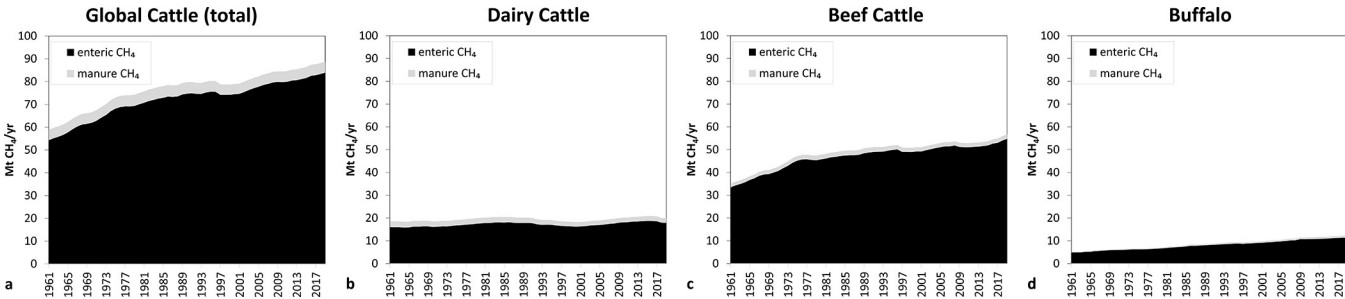

**Fig 2.** Methane emissions from global cattle (aggregating dairy and beef cattle and buffalo) (a), dairy cattle (b), beef cattle (c) and buffalo (d) for both enteric and manure sources. Results are expressed as Mt $CH_4$ per year. Data source: FAOSTAT.

cumulative $CO_2$ emissions to obtain an estimate of temperature change due to the $CO_2$ burden experienced [15]. According to IPCC (2021) [1], each Tt of cumulative $CO_2$ emissions is assessed to likely cause a 0.27˚C to 0.63˚C increase in global surface temperature with a best estimate of 0.45˚C.

For other non-$CO_2$ GHGs, a relationship between the cumulative amount emitted and their induced global warming can only be approximated if the GHG is a long-lived gas such as $CO_2$ (e.g. nitrous oxide: $N_2O$) and when the gas is expressed as $CO_2$-e using the $GWP_{100}$ metric [13]. For GHGs with short-life in the atmosphere (i.e. $CH_4$), a new climate metric was developed, denoted GWP* with the associated $CO_2$ warming-equivalent ($CO_2$-we), to provide a direct link between calculated cumulative $CO_2$-we emissions and global warming via the TCRE value [16].

The GWP* metric was applied in this study to calculate $CO_2$-we emissions for $CH_4$ sources from livestock and global cattle. The following equation, developed by Smith et al. [17] that is adaptable to calculate $CO_2$-we for all short-lived GHG on a particular year, was used:

$$CO2we = 4.53 \times E100(t) - 4.25 \times E100(t - 20) \tag{1}$$

Where, E100 corresponds to conventional global warming $CO_2$-equivalent emissions calculated using $GWP_{100}$. This value is required for both the emission rates for the current year, $E100$ $(t)$, and from 20 years ago, $E100$ $(t–20)$. When there is a large difference between these two emission rates, a large $CO_2$-we value is returned, emphasizing the significant and rapid impact of changing methane emission rates.

Emissions are used from years t (the year for which $CO_2$-we emissions are being calculated) and from t-20 (20 years prior). This allows GWP* to represent the impact of new emissions (which cause strong additional warming), stable emissions (minimal additional warming) and reducing emissions (which reverses some past warming) [6].

Methane emissions were converted into $CO_2$-e using the value indicated by the IPCC report AR6 of $GWP_{100}$ = 27.2 ([1]).

The following steps and calculations were performed to estimate the induced global warming (global temperature change) by the different global emissions sources from a particular period:

i. Selection of emissions time series which is defined by an initial year (e.g. 1750, 1961, or 2020) and an ending year (e.g. 2019, 2050, or 2100).

ii. Calculate the cumulative emissions across the study period. For $CH_4$, it is required that emissions are expressed as $CO_2$-we emissions following Eq 1. Although not applicable in our study, other non-$CO_2$ GHG sources like $N_2O$ would require that emissions are

expressed as $CO_2$-e emissions using $GWP_{100}$. For $CO_2$ emission sources, no conversion is required.

iii. Multiply the value of cumulative emissions by the TCRE value to obtain global warming induced in the study period by these emissions. For this study, although the TRCE value of 0.45 K˚/Tt $CO_2$ [1] is used throughout all scenarios, an alternative value of 0.4 K˚/Tt $CO_2$ (used by Lynch et al. [8]) is used for the long-term historical livestock $CH_4$ emissions series as an illustration of the sensitivity of the final global warming result to changes in the choice of the TRCE value.

## Scenarios for testing the framework to estimate the impact of GHG emissions on global temperature change

Different scenarios were developed and shown in Table 1 to illustrate the usage of GWP* for retrospective time-series analysis (1–3) and for forecasting future contributions to warming from different emission sources (4–5):

1. Comparison between global warming induced by long-term (1750–2019) historical emissions of fossil $CO_2$, LUC $CO_2$, and livestock $CH_4$ according to the following assumptions:

    i. no livestock emissions during 1750–1859 (as assumed by Reisinger and Clark (2018) [12])

    ii. annual livestock emissions in 1750 were half of 1860 considering that the global human population was approximately half and had, and increased gradually up to 1860 (this scenario also assumes a non-changing meat and milk consumption per capita)

    iii. annual livestock emissions were similar to 1860 during the 1750–1860 period.

It must be noted that, the 'global warming induced' calculated in this scenario reports the global temperature change since pre-industrial years or, in other words, the absolute level of global warming attributable to these sectors and assumed as anthropogenic nowadays. By including a period that goes back to preindustrial time, we could explore warming impacts relative to a 'reference condition' of no anthropogenic emissions (marginal warming). Setting the reference time in this scenario to preindustrial times reflects the type of information provided by the $GWP_{100}$ (marginal warming-basis), but retains a dynamic component in the case of GWP* [18]. The difference between marginal and 'additional' warming, as measured by GWP* in the manner it is mostly applied, is explained in detail in the IPCC WGIII contribution to the AR6 Report, particularly in supplementary material to chapter 2 ([1]).

**Table 1. Overview of the set-up of the different scenarios for testing the framework to estimate the impact of GHG emissions on global temperature change.**

| Scenario | Analysis type | length of period | warming basis | reference year (s) | target year | Sectors involved |
|---|---|---|---|---|---|---|
| 1 | retrospective (past) | long-term | marginal | 1750 | 2019 | livestock $CH_4$, fossil $CO_2$, LUC $CO_2$ |
| 2 | retrospective (past) | short-term | additional | 1981, 1990, 2000, 2010 | 2019 | Global cattle $CH_4$, fossil $CO_2$, LUC $CO_2$ |
| 3 | retrospective (past) | short-term | additional | 1981, 1990, 2000, 2010 | 2019 | Global cattle $CH_4$ |
| 4 | forecasting (future) | long-term | marginal | 1750 | 2100 | livestock $CH_4$, fossil $CO_2$, LUC $CO_2$ |
| 5 | forecasting (future) | short-term | additional | 1990 | 2050 | Global cattle $CH_4$ |

'Marginal warming' and 'Additional warming' denote the warming that emissions causes, relative to the absence of that emission and a pre-existing level of emissions at the reference year, respectively [1,18].

2. Comparison between global warming induced by short-term (1961–2019) historical emissions of fossil $CO_2$, LUC $CO_2$, and global cattle $CH_4$ for the following periods: 1981–2019, 1990–2019, 2000–2019, and 2010–2019.

It must be noted that, the 'global warming induced' calculated as in this scenario, reports the temperature change over time, relative to a reference level of warming caused by prior emissions up to the beginning of the time series being evaluated.

3. Comparison between cumulative $CO_2$ equivalent emissions expressed as $CO_2$-we (using GWP*) vs. $CO_2$-e (using $GWP_{100}$) for short-term (1961–2019) historical global cattle $CH_4$ emissions for the following periods: 1981–2019, 1990–2019, 2000–2019, 2010–2019.

This comparison will be useful to estimate the contribution of $CH_4$ emission rates affecting the remaining carbon budget (as expressed by cumulative $CO_2$-we emissions) and hence, the error that would imply expressing $CH_4$ as cumulative $CO_2$-e emissions in C budget calculations. The C budget is defined as the cumulative amount of $CO_2$ emissions, up to net-zero, that would be consistent with limiting warming to a specified level while considering the contribution of non-$CO_2$ climate forcers to total warming [19].

4. Comparison of projected global warming associated with future emission pathways for global fossil $CO_2$, LUC $CO_2$, and livestock $CH_4$ emissions (2020–2100 period) for 3 different emission rates assumptions:

    i. keeping emissions unchanged from 2020

   ii. reducing global GHG emissions in a sustained way (1% decrease per year, constant over time)

   iii. reducing global GHG emissions up to reaching 0 emissions (net-0 CO2 or net-0 $CO_2$-e using $GWP_{100}$ for livestock $CH_4$) by 2100 by reducing fossil $CO_2$: 0.44 Gt $CO_2$/yr; LUC $CO_2$: 0.04 Gt $CO_2$/yr and livestock $CH_4$: 1.5 Mt $CH_4$/yr.

In these projected scenarios, warming associated to future emission pathways will include the estimated warming legacy from 1750. Hence, warming results will refer to absolute temperature levels attributable to each sector since the industrial revolution started.

5. Analyzing the projected global warming associated with 3 futures global $CH_4$ cattle emission pathways leading to a stabilisation of global temperatures at year 2050 (1990–2050 period) varying in the time and intensity when absolute $CH_4$ emission rates are reduced as follows:

    i. fast: large reductions (greater than those required for sustained emission rate reductions: 0.5% decrease per year) in global cattle $CH_4$ emissions are introduced during the first decades (2020–2040) followed by reductions at smaller rates (2040–2050) (0.1% decrease per year).

   ii. sustained: reductions in global cattle $CH_4$ emissions in a sustained way starting in 2020 (0.32% decrease per year, constant over time)

   iii. delayed: increase in absolute emission rates for the first 2 decades (2020–2040) (0.25% increase per year), followed by large reductions (0.84% decrease per year)

in global cattle $CH_4$ emissions are introduced during the last decade (2040–2050).

Temperature stabilisation is achieved in the year 2050 when $CH_4$ emissions expressed as $CO_2$-we using GWP* reach the value of zero (0). It must be noted that reaching zero (0) $CO_2$-we emissions at a particular year can be assumed analogous to the concept of stabilizing

temperatures by reducing $CO_2$ emissions when reaching net-zero $CO_2$ emissions [20]. Aggressive $CH_4$ mitigation could reverse much of the $CH_4$-induced warming experienced at a particular year, having an analogous temperature impact to actively removing past $CO_2$ emissions (e.g. by reforesting), and hence reported as a negative $CO_2$-we (i.e. $CO_2$-we/yr < 0). It must be noted that the 'global warming' calculated in these scenarios reports the temperature change over time relative to a reference level of warming caused by prior emissions up to the beginning of the time series being evaluated.

## Results and discussion

### Global warming induced by long-term (1750–2019) historical emissions of fossil fuel $CO_2$, LUC $CO_2$, and livestock $CH_4$

Our results show that the estimated global warming associated with fossil fuel $CO_2$, LUC $CO_2$, and livestock $CH_4$ cumulative emissions from 1750 to 2019 were 0.75°C, 0.33°C and 0.15°C, respectively (Fig 3A). This represents total global warming of 1.23°C corresponding to cumulative emissions of 1663 Gt $CO_2$ from fossil fuel, 737 Gt $CO_2$ from LUC, and 9.5 Gt $CH_4$ from livestock that equals 258 Gt $CO_2$-e when using $GWP_{100}$ and 333 Gt $CO_2$-we when using GWP* in the calculations. As expected, using the lower TRCE value of 0.4 K°/Tt $CO_2$ (compared with 0.45) results in a proportional reduction in estimated global warming values to 0.67°C, 0.29°C and 0.13°C for fossil fuel $CO_2$, LUC $CO_2$, and livestock $CH_4$ emissions, respectively (Fig 3B). The total global warming estimated in this study for the 1750–2019 period (1.23°C) is comparable to total human-caused global surface temperature increases from 1850–1900 to 2010–2019 of 0.8°C-1.3°C reported by IPCC (2021) [1]. Our estimates for global livestock-induced warming (0.15°C) are slightly greater than those reported (0.11°C) using the MAGICC climate model for the slightly shorter period 1750–2010 [12].

Considering that livestock existed for many centuries before the industrial revolution began, the scenarios (1.ii and 1.iii) that included livestock emissions between 1750 and 1859 led to lower historical contributions to warming (0.12–0.13°C, Fig 3A) than the zero livestock emissions scenario (1.i). When livestock emissions were set to zero prior to 1860, the GWP*-TCRE calculation estimates warming on a marginal-basis, in other words the temperature changes are compared to those emissions not occurring. In this sense, using the GWP*-TCRE

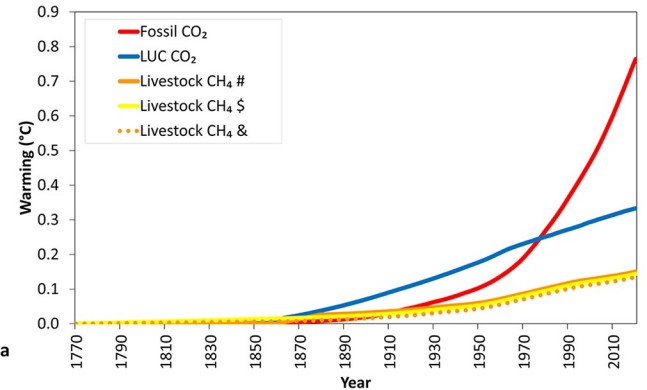
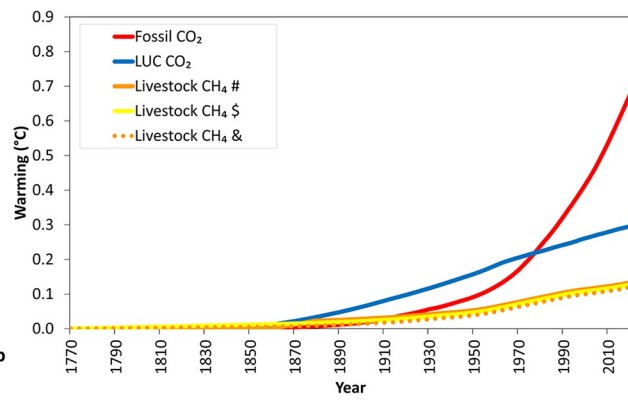

**Fig 3.** Historical warming impact of global fossil $CO_2$ emissions excluding carbonation (Fossil $CO_2$), global land use change (LUC $CO_2$) emissions and livestock $CH_4$ as estimated by multiplying aggregated $CO_2$ emissions and $CO_2$-we (using GWP*) (for $CH_4$ sources) emissions by the TCRE value: (a) 0.45°K/Tg $CO_2$ and (b) 0.40°K/Tg $CO_2$. For livestock, there are 3 different assumptions of $CH_4$ emission rates from global livestock in the period between 1750–1860: # and solid orange line: 0 $CH_4$/yr emissions, $ and yellow line: Sustained increased emissions from 11.5 to 22.3 kt $CH_4$/yr and & and dotted orange line: 22.3 kt $CH_4$/yr during 1750–1860.

**Table 2. Additional temperature from fossil $CO_2$ emissions excluding carbonation, land use change (LUC) $CO_2$ and livestock $CH_4$ for different periods (1770–1970; 1970–1990; 1990–2019) and % of this temperature within each period that can be attributed to each sector.**

|  | Additional temperature (˚C) | Fossil $CO_2$ | LUC $CO_2$ | Livestock $CH_4$ |
|---|---|---|---|---|
| **1770–1970** | 0.51 | 37% | 45% | 17% |
| **1970–1990** | 0.24 | 71% | 17% | 12% |
| **1990–2019** | 0.49 | 81% | 12% | 7% |

calculation on those scenarios that reflect livestock $CH_4$ emissions prior to 1750, will allow us to consider the level of $CH_4$ emissions already existing in 1750 (prior to the reference time we generally account for the antropogenic emissions in the IPCC-based frameworks) and consequently, allow us to adjust the warming that livestock $CH_4$ has added since the industrial revoluition (reference time). At the country level, although a comparison of the current anthropogenic livestock $CH_4$-induced warming with the pre-industrial reference will generally show a large net warming due to much higher current $CH_4$ emission for some specific countries (e.g. Germany). $CH_4$ estimates suggest that, compared with XIX century, Germany's livestock has been emitting less enteric $CH_4$ since 2003 [21].

Table 2 shows the estimated additional warming attributed to global $CO_2$ (fossil and land use change) and livestock $CH_4$ for three different periods: 1770–1970, 1970–1990, and 1990–2019. The additional warming contributed by all 3 sources during the first 200 years (0.51˚C for 1770–1970) is similar to the additional warming contributed during the last 3 decades (0.49˚C for 1970–2019). However, $CO_2$ from fossil sources contributes the vast majority, at more than 80% of the of the total warming over the past three decades. In fact, the relative contribution of fossil fuel emissions to global warming accelerated over time while those from LUC and global livestock decreased over time with Sanderman et al. [22] reporting similar findings. In contrast, the contribution by LUC $CO_2$ was particularly severe prior to 1970 accounting for 45% of the additional warming associated with these three sources of global emissions. The contribution to global warming by livestock $CH_4$ represented 17% from the three sources examined up to 1970 and shrunk to 7% in the last three decades (1990–2019). The historical contributions to warming from each source are quite different. LUC contributions peaked by 1970 and are now decreasing. Fossil fuel contributions always increased since the start of the industrial revolution and accelerated dramatically since the mid-20th century. Livestock contributions increased throughout the period studied but at a much lower rate than fossil fuel contributions.

## Global warming induced by short-term (1961–2019) historical emissions of fossil fuel $CO_2$, LUC $CO_2$, and global cattle $CH_4$

Short-term global warming estimates associated with global cattle $CH_4$ emissions (dairy cattle, beef cattle and buffalo) were 0.028˚C for 1981–2019 and 0.019˚C for 1990–2019 (Fig 4). For both periods, when animal types are dissaggregated, we found that the largest additional global warming can be attributed to enteric $CH_4$ from the beef cattle sector followed by enteric $CH_4$ from the buffalo and dairy cattle sectors (Fig 4).

Fig 5 shows that the contribution of emissions from fossil fuel to historical warming from 1981 to 2019 is 12-fold greater than from global cattle $CH_4$ and 6-fold greater than from LUC. Also, the additional warming per year associated to global cattle $CH_4$ during this period is in the range of 4 to 6% compared with the additional warming attributed to fossil fuel $CO_2$ emissions. This is a significant difference that provides perspective on the contributions from the three different GHG emission sources. Moreover, the averaged annual additional warming

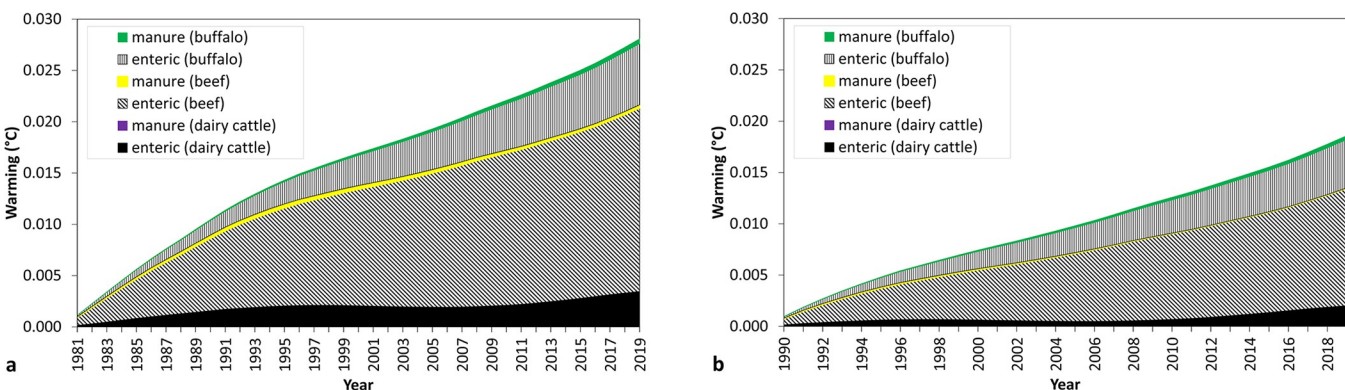

**Fig 4.** Additional warming associated with methane emissions from different sources (enteric and manure) of global cattle systems (dairy, beef and buffalo) until 2019 and since 1981(a) and 1990 (b). Additional warming is calculated by multiplying cumulative $CO_2$-we (using GWP*) emissions with the TCRE value (0.45˚K/Tg $CO_2$-we).

associated to fossil $CO_2$ emissions, which expresses the relative annual added temperature (as calculated by dividing the additional warming within the analysed period by the number of years in such period), became greater in more recent years (from 12 m˚C/yr in 1981–2019 to 16 m˚C/yr in 2010–2019).

Although not explored in this study, when the approach is applied to emission assessments at sub-global scale (e.g. country-level: [23–25]), its applicability, similarly as with $CO_2$ ([16]),

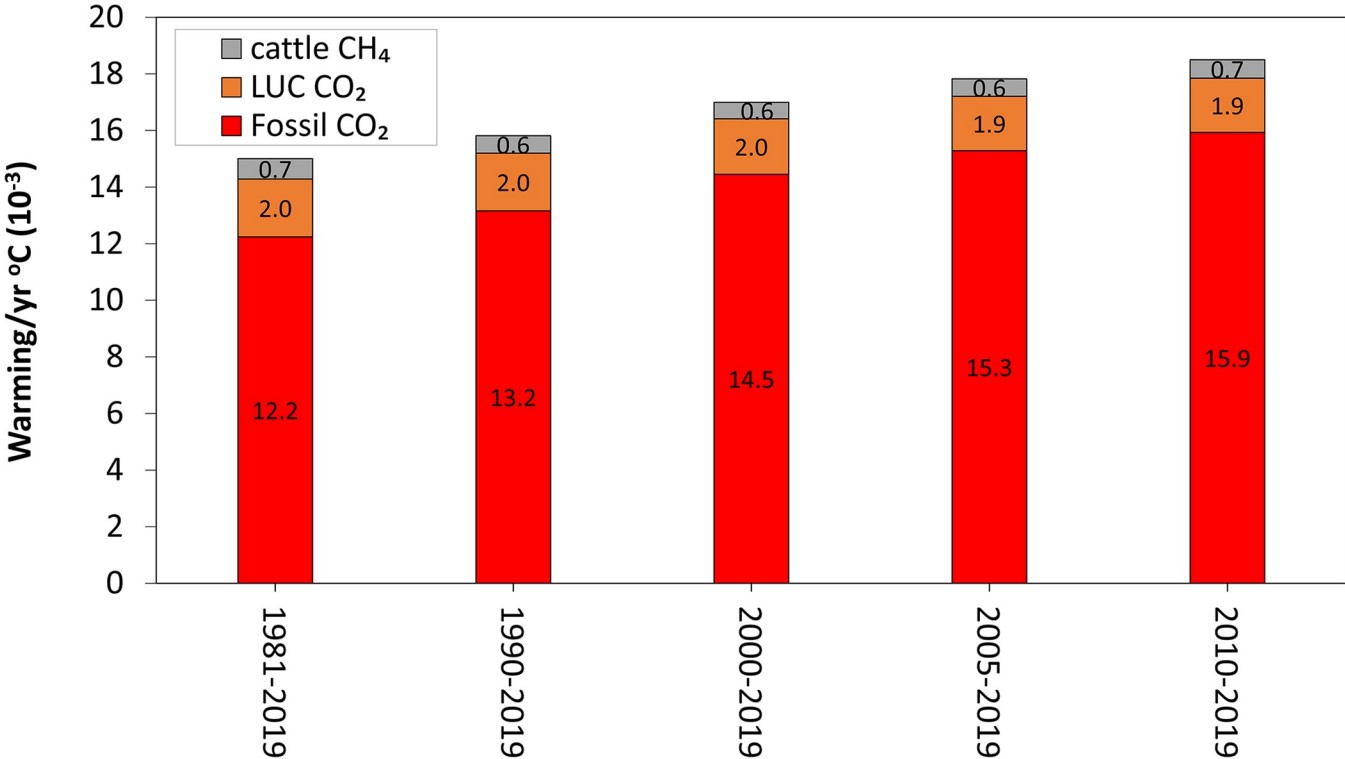

**Fig 5.** Averaged annual additional warming (as calculated by dividing the additional warming within the analysed period by the number of years in such period) associated with global fossil fuel $CO_2$, land use change $CO_2$ and global cattle $CH_4$ for periods with different reference years (1981, 1990, 2000, 2005 and 2010) until 2019.

should carefully consider potential equity impacts ([26]). For example, Reisinger et al. [19] suggest having a clear separation of legacy warming from past emissions, which is significant only for long-lived gases, and marginal warming from current and future emissions and removals, which applies for all gases, to best address mitigation strategies. Nevertheless, how sub-global scale entities, such as countries, might incorporate the warming from past $CO_2$ or $CH_4$ emissions into climate policy poses questions that go beyond the scope of this study.

## Comparison between cumulative $CO_2$ equivalent emissions expressed as $CO_2$-we (using GWP*) vs. $CO_2$-e (using $GWP_{100}$) for short-term (1961–2019) historical global cattle $CH_4$ emissions for the following periods: 1981–2019, 1990–2019, 2000–2019, 2010–2019

Cumulative (Fig 6A) and average annual emissions ($CO_2$ equivalent emissions per year) expressed as $CO_2$-we (using GWP*) and $CO_2$-e emissions (using $GWP_{100}$) associated to global cattle $CH_4$ emissions varied depending on length of period assessed. As expected, cumulative emissions expressed as $CO_2$-e increase linearly with longer assessment periods, but when expressed as $CO_2$-we, the changes in cumulative emissions responded to the rate of change of $CH_4$ emissions. Therefore, cumulative $CO_2$-we values were lower than if reported as $CO_2$-e by 38% on average across all periods examined (Fig 6A), but the differences ranged from 28% (1981–2019) to 44% (2000–2019) depending on how $CH_4$ emission rates changed in the years included in each period. This observation implies that each tonne of global cattle $CH_4$ emissions contributed less additional warming than that from 27.2 tonnes of any source of $CO_2$ (27.2 is methane's $GWP_{100}$ value) in all the periods examined. In fact, increasing the rate of $CH_4$ emissions has a greater impact on global mean surface temperature per tonne of $CH_4$ emitted than constant $CH_4$ emissions, while $CO_2$ emissions have the same impact on global mean surface temperature per tonne of $CO_2$ emitted regardless of emission trajectory [6].

Three other studies that compared $CH_4$ emission pathways expressed as $CO_2$-e using $GWP_{100}$ and $CO_2$-we using GWP* found similar results. Del Prado et al. [27] estimated additional warming associated to historical direct GHG emissions from dairy small ruminants in continental Europe. These authors report that the European sheep and goat dairy sector did not contribute to additional warming in the 1990–2018 period, while also reporting larger cumulative emissions expressed as $CO_2$-e than expressed as $CO_2$-we. At the country level,

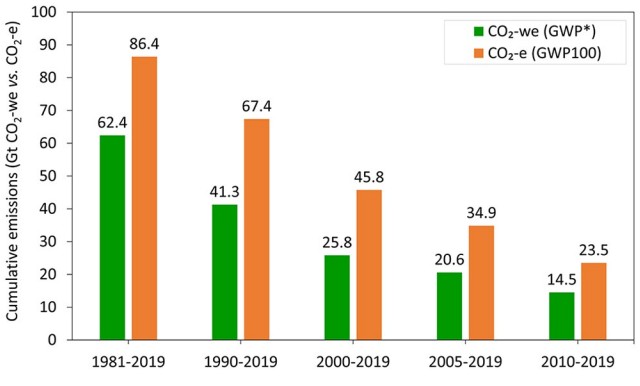
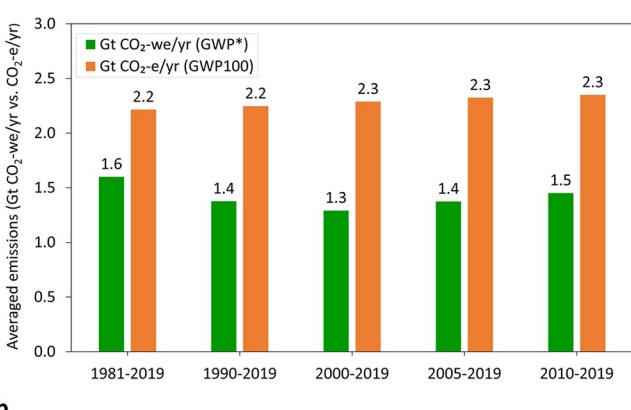

**Fig 6.** Cumulative emissions (a) and average annual emissions (as calculated by dividing the cumulative emissions within the analysed period by the number of years in such period) (b) as expressed using GWP* ($CO_2$-we) and conventional $GWP_{100}$ ($CO_2$-e) coming from methane emissions from global cattle systems for different reference years (1981, 1990, 2000, 2005 and 2010) until 2019. (white: $CO_2$-we and grey: $CO_2$-e). Note that axes from Fig 6A and 6B have different magnitudes.

Gilreath et al. [25] also found that historical (1920–2020) enteric $CH_4$ emissions from US beef production were larger when expressed as $CO_2$-e than $CO_2$-we. Moreover, whereas differences between cumulative $CH_4$ emissions were in the 9 to 30% range depending on the level of methodological complexity used (also denoted as tier level by IPCC (2019) [28]), differences between annual emission rates were larger than that for any given year. Finally, Hörtenhuber et al. [24] found a large discrepancy between the climate impact from historical (2005–2019) GHG emissions from Austrian livestock estimated using $GWP_{100}$ and GWP* since annual $CH_4$ emission rates decreased over the period examined.

## Projected global warming associated with future emission pathways for global fossil $CO_2$, LUC $CO_2$, and livestock $CH_4$ emissions (2020–2100 period)

Fig 7 shows estimations of warming impacts of future emissions pathways (2020–2100) considering the legacy of warming since the industrial revolution (1770–2019) for 2 global sources of anthropogenic $CO_2$ emissions (fossil and LUC) and $CH_4$ from enteric fermentation and manure from livestock. Whereas for both $CO_2$ sources all pathways, unchanged emissions, reducing emissions at 1% per annum (sustained reduction, constant over time), and reducing emissions to net zero by 2100, result in increased global warming above 2019 levels, only unchanged emissions lead to increased global warming above the 2019 level for $CH_4$.

To put this in context, our estimates for historical warming until 2019 associated to $CO_2$ from fossil (0.75˚C), LUC (0.33˚C), and livestock $CH_4$ (0.15˚C) amounts to 1.23˚C for the 3 sources (data in Supplementary Material), which implies that 61%, 27% and 12% of this warming can be attributed to fossil $CO_2$, LUC $CO_2$, and livestock $CH_4$ emissions, respectively. Keeping these sources of emissions unchanged from 2019 until 2100 implies that the relative contribution of fossil $CO_2$ to global warming would be much greater (76%) than LUC $CO_2$ (17%) and livestock $CH_4$ (7%) by 2100. Moreover, keeping these emissions unchanged would result in global warming contributions increasing by 169% (from 0.75˚C to 1.38˚C), 36% (from 0.33˚C to 0.45˚C), and 27% (from 0.15˚C to 0.19˚C) from 2019 levels for fossil $CO_2$, LUC $CO_2$, and livestock $CH_4$, respectively.

Reducing emissions by 1% per annum, constant over time, further reduces the relative contribution to global temperatures above the pre-industrial period of livestock $CH_4$ emissions (0.12˚C: 6%) compared with fossil $CO_2$ (1.62˚C, 75%) and LUC $CO_2$ (0.41˚C, 19%). In fact,

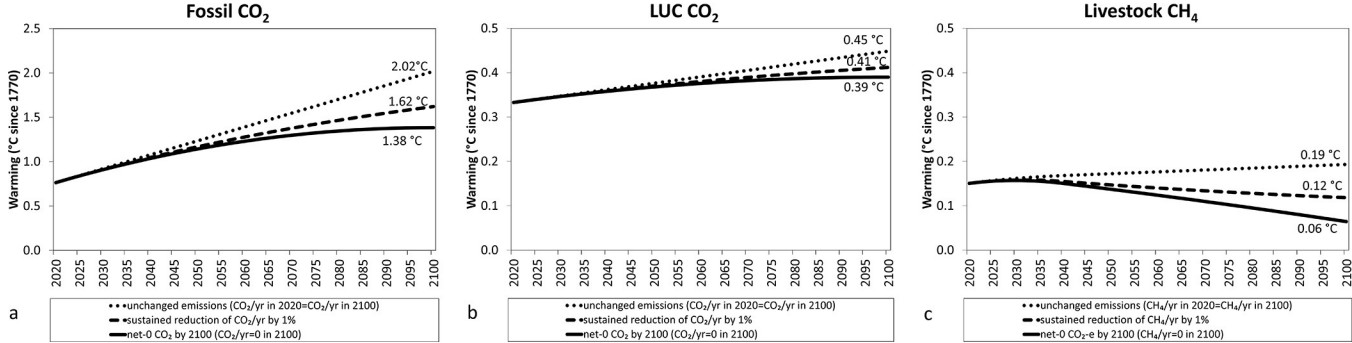

**Fig 7.** Warming impact of global fossil $CO_2$ emissions (a), global land use change (LUC) emissions (b) and livestock $CH_4$ (c) as estimated by multiplying aggregated $CO_2$ emissions and $CO_2$-we (using GWP* for $CH_4$) emissions by the TCRE value (0.45˚K/Tg $CO_2$) since 1770 and considering 3 different future pathways of emissions (2020–2100) (unchanged, reducing 1% per annum, constant over time and reducing emissions gradually until reaching 0 emissions (net-0 $CO_2$ or $CO_2$-e using $GWP_{100}$ for livestock $CH_4$) in 2100: Reducing fossil $CO_2$: 0.44 Gt $CO_2$/yr; LUC $CO_2$: 0.04 Gt $CO_2$/yr and livestock $CH_4$: 1.5 Mt $CH_4$/yr). Note that axes for global fossil $CO_2$ is x5 times larger than that for LUC $CO_2$ and livestock $CH_4$.

reducing livestock $CH_4$ emissions by 1% per annum by 2100, would decrease their impact on global warming to levels similar to the early 1990s or a 20% reduction compared with warming levels at 2019 (0.12˚C *vs.* 0.15˚C). In contrast, a yearly decrease of 1% in $CO_2$ emissions would still result in an increase in warming for the year 2100 compared to the existing warming caused in 2019. The warming associated with fossil fuel emissions would increase by 116%, while the warming linked to LUC emissions would rise by 24%.

Reducing all emission rates to zero by 2100 (i.e. $CO_2$/yr = 0 or $CO_2$-e = 0 for livestock $CH_4$) would lead to reversing livestock $CH_4$ emissions warming impacts in 2100 to levels observed in the 1940s-1950s, such as a 0.06˚C increase relative to 1750 or a 60% reduction relative to 2019. Those warming impacts from livestock $CH_4$ emissions would represent 3% of the total warming attributed to the three sources by 2100. Meanwhile, pathways leading to net zero $CO_2$ emissions from fossil fuel or LUC would lead to stabilising their impact on global temperatures at 1.38˚C (84% relative to 2019 year) and 0.39˚C (18% relative to 2019 year) above pre-industrial temperatures, respectively. Unlike $CO_2$ where it is currently technically possible to reduce net emissions to zero, most technologies for $CH_4$ removals are still under development [29]. This limitation for reaching net zero $CH_4$ emissions is reflected in the pathways simulated by integrated climate-economic modelling to limit global temperature to 1.5 ˚C above pre-industrial times, where biogenic $CH_4$ is reduced by 24% to 47% from unspecified sectors relative to 2010, while $CO_2$ must be reduced to net-zero ([30]).

These reductions in livestock $CH_4$ emissions are also viewed as necessary by Reisinger et al. [19], who argue that future livestock $CH_4$ emissions significantly constrain the remaining C budget and the ability to meet stringent temperature limits (i.e. 1.5˚ C goal). Reisinger et al. [19] advocated for strategies to reduce $CH_4$ emissions through more efficient production, technological advances and demand side changes, and importantly always carefully considering their interactions with land-based C sequestration. The challenge is that projected growth in livestock production systems and current $CH_4$ emissions from ruminant livestock are greatest in low-and middle-income countries where the nutrient supply to the population could be severely compromised if animal-derived food demand is dramatically shortened. In addition, most production-side mitigation strategies that were designed for developed countries don't apply to the production systems used in low- and middle-income countries [31,32]. These socioeconomic and environmental aspects of mitigation were ignored in some studies [33] that explored the climate impact of implementing even more drastic scenarios, such as eliminating animal agriculture. Meanwhile, other studies went beyond the livestock sector and explored pathways for global food systems achieving net-zero emissions by 2050 using GWP* [20] or studied the implications on optimal mitigation options of choosing a particular climate metric for $CH_4$'s warming potential [34].

## Projected global warming associated with future global cattle $CH_4$ emission pathways leading to a stabilisation of global temperatures by 2050

Additional warming is stabilised by 2050 at 0.026˚C relative to the temperature on 1990 when global cattle $CH_4$ emissions are reduced at a sustained annual rate of 0.32% starting in 2020 (Fig 8). This means that $CH_4$ emissions greater than a 0.32% annual decrease will contribute to additional warming in the future. This result is consistent with those obtained when applying GWP* to global livestock [20] and US beef [23] $CH_4$ emissions pathways and those presented by the GWP* original study [7] where additional warming computed from $CO_2$-we was compared with that obtained using a simple climate model.

The sustained decrease in annual global cattle $CH_4$ emissions of 0.32% is equivalent to a reduction of annual $CO_2$-we emissions from 1.8 Gt in 2019 to 0 in 2050 (green dotted line).

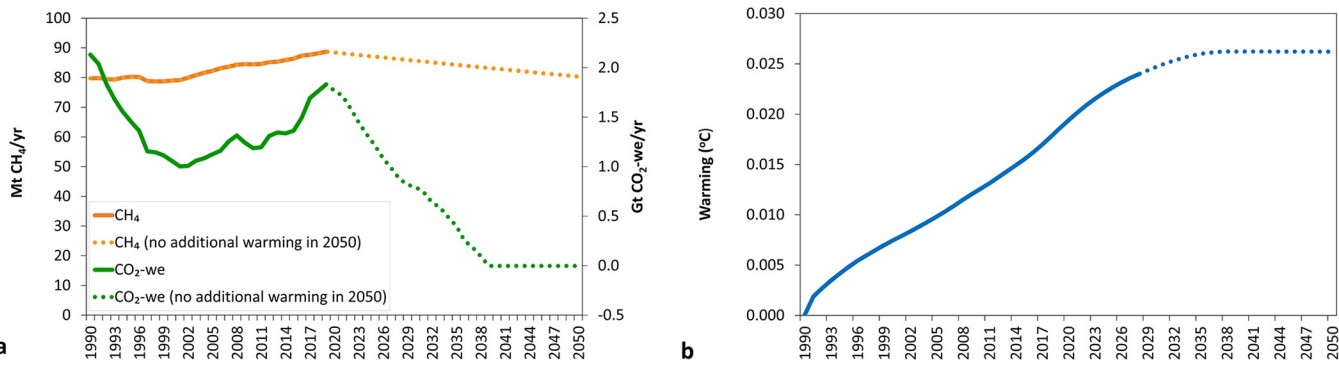

**Fig 8.** Historical (1990–2019: solid line) and future (2020–2050: dotted line) pathways for global cattle $CH_4$ emissions leading to an stabilisation or no additional warming of their impact on global temperature (both historical and future expressed as $CH_4$: green line or $CO_2$-we using GWP*: orange line) (a) and its corresponding impact on additional temperature caused (blue line, historical: solid line, future: dotted line) as estimated by multiplying aggregated $CO_2$-we (using GWP*) emissions by the TCRE value (0.45˚K/Tg $CO_2$) (b).

These $CH_4$ emissions from global cattle would cause additional warming since the reference year (1990) comparable to that from a source of $CO_2$ at the rates that are expressed as $CO_2$-we. In this case, it would be similar to the emissions of net-zero $CO_2$, as depicted by the green line in Fig 8. Altogether, achieving a cumulative reduction of 9.2% by 2050, compared to the year 2020, is necessary for $CH_4$ emission rates. This would entail reducing annual emissions from approximately 89 Mt $CH_4$/yr (in 2020) to approximately 80 Mt $CH_4$/yr by 2050, as indicated by the orange dotted line in Fig 8. Other studies that investigated the impact of different GHG pathways in the livestock sector found dimilar results. Del Prado et al. [27] reported that the European sheep and goat dairy sector could reverse all the warming caused from 2020–2100 to 2020 levels by keeping its production at the level of 2020 and introducing $CH_4$ mitigation and C sequestration measures. In another study, Liu et al. [35] showed that the California dairy industry would not contribute additional warming compared to the levels of 1970 in 2030 if $CH_4$ emissions can be reduced by 1% per year.

Fig 9 compares the global temperature change during the 1990–2050 period by three modelled emission pathways for global cattle $CH_4$ emissions that result in no additional warming by 2050 (i.e. $CO_2$-we = 0 at the target year of 2050). The three $CH_4$ emission reductions pathways as explained in more detail in Materials and Methods section are: 1) 'fast' mitigation introduced in 2020, 2) 'sustained' mitigation introduced with a fixed reduction rate per year as in Fig 8 (0.32% reduction per year, constant over time), and 3) a 'delayed' mitigation strategy introduced in the year 2040 and onwards. Although all three modelled emission pathways lead to stabilising the additional warming caused by $CH_4$ emissions by 2050, the pathways that mitigated $CH_4$ earlier stabilize warming at a lower (0.024˚C) temperature than those that delay $CH_4$ mitigation (0.035˚C). Del Prado et al. [27] found, for the context of European dairy small ruminants' systems, that the speed of introduction of mitigation measures makes a considerable near-term impact but a smaller difference by end of century. Essentially, it is not only about stabilising the impact of emissions on additional warming (i.e. reaching zero $CO_2$-we), but the rate of change and speed in which $CH_4$ mitigation occurs in the reduction pathway that determines the total warming contribution by global cattle $CH_4$ emissions by the target date.

## Conclusions

Using the GWP* to express historical global livestock $CH_4$ emissions as cumulative $CO_2$-we that are converted to temperature change by multiplication with TRCE allows the comparison

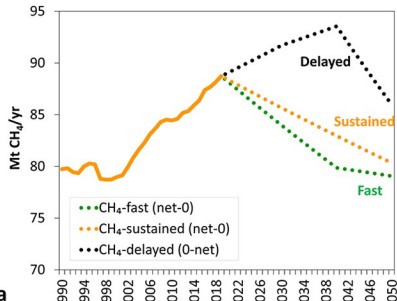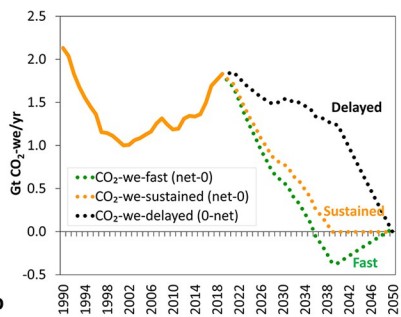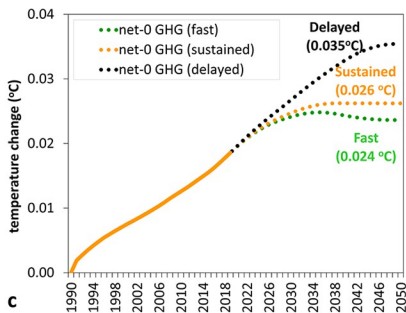

**Fig 9.** Historical (1990–2019: solid line) and 3 future (2020–2050: dotted line) reduction pathways for global cattle $CH_4$ emissions, all leading to a stabilisation of their impact on global temperature (a). These emissions are expressed as $CO_2$-we (warming-eq emissions) using GWP* (b) and its corresponding impact on additional temperature caused is estimated by multiplying aggregated $CO_2$-we (using GWP*) emissions by the TCRE value (0.45˚K/Tg $CO_2$) (c). The three future pathways correspond to a 'fast' first $CH_4$ reductions (dotted green line), 'sustained' reductions at 0.32% decrease per year (constant over time) (dotted orange line) and 'delayed' introduction of $CH_4$ reductions (dotted black line).

of global warming emission pathways in an accurate but simpler way than using a climate model [36]. Clearly reductions in global cattle $CH_4$ emission rates are needed for their contributions to stop additional warming. A sustained annual reduction of 0.32% would be sufficient to stabilise the effect on global temperatures by global cattle $CH_4$ emissions. Conversely, $N_2O$ and $CO_2$ emissions must be eliminated, or equivalent amounts actively removed from the atmosphere (zero emissions), to obtain the same effect on global warming. In fact, as seen with the example for global cattle, reductions in $CH_4$ emissions above 0.32% annually would cause analogous effects as increasing C sinks (i.e. active removal of long-lived GHG) and larger reductions would reverse historical temperature impacts from previous decades. As detailed in this study, applying key interventions such as those analysed in [37] to reduce global cattle $CH_4$ can even lead to stabilising global temperatures, but the peak temperature achieved would be different and dependent on how quickly and aggressive are the mitigation strategies applied. Moreover, reductions greater than 0.32% annually would partially undo past contributions by global livestock and global cattle to temperature increases, be analogous to active $CO_2$ removal from the atmosphere and, depending on the extent of the reduction, approximate to net-zero GHG as defined using $GWP_{100}$.

## Supporting information

**S1 File. Spreadsheet with all the data used for the figures.**
(XLSX)

## Acknowledgments

Many thanks to Arla Foods, Dairy Australia, Dairy Companies of New Zealand, Dairy Management Inc., Global Dairy Platform, Global Round Table for Sustainable Beef, McDonalds Corporation, and Meat and Livestock Australia for helping on the study design and providing some of the general questions.

## Author Contributions

**Conceptualization:** Agustin del Prado, Brian Lindsay, Juan Tricarico.

**Data curation:** Agustin del Prado, Juan Tricarico.

**Formal analysis:** Agustin del Prado, Juan Tricarico.

**Funding acquisition:** Brian Lindsay, Juan Tricarico.

**Investigation:** Agustin del Prado, Brian Lindsay, Juan Tricarico.

**Methodology:** Agustin del Prado, Juan Tricarico.

**Project administration:** Brian Lindsay.

**Resources:** Brian Lindsay.

**Software:** Agustin del Prado.

**Supervision:** Juan Tricarico.

**Validation:** Brian Lindsay, Juan Tricarico.

**Visualization:** Agustin del Prado.

**Writing – original draft:** Agustin del Prado.

**Writing – review & editing:** Brian Lindsay, Juan Tricarico.

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
