## [Decision Letter · Decision Letter 0]

26 Apr 2023

PONE-D-23-08709Retrospective and projected warming-equivalent emissions from global livestock and cattle calculated with an alternative usage of global warming potentials denoted GWP*PLOS ONE

Dear Dr. del Prado,

Thank you for submitting your manuscript to PLOS ONE. After careful consideration, we feel that it has merit but does not fully meet PLOS ONE’s publication criteria as it currently stands. Therefore, we invite you to submit a revised version of the manuscript that addresses the points raised during the review process.

We look forward to receiving your revised manuscript.

Kind regards,

Zhaoxia Guo

Academic Editor

PLOS ONE

Journal Requirements:

   "This research was partially supported by contributions from (in alphabetical order) Arla Foods, Dairy Australia, Dairy Companies of New Zealand, Dairy Management Inc., Global Dairy Platform, Global Round Table for Sustainable Beef, McDonalds Corporation, and Meat and Livestock Australia. BC3 is supported by the Basque Government through the BERC 2022-2025 program and by Spanish Ministry of Economy and Competitiveness MINECO through BC3 María de Maeztu excellence accreditation 2018–2022 (Ref. MDM-2017-0714), funded by MCIN/AEI/10.13039/501100011033/. Agustin del Prado is financed by the programme Ramon y Cajal from the Spanish Ministry of Economy, Industry and Competitiveness (RYC-2017-22143) and Ikerbasque."

  "AdP was supported in this study specifically by the Global Dairy Platform (GDP). AdP receive funds by the programme Ramon y Cajal from the Spanish Ministry of Economy, Industry and Competitiveness (RYC-2017-22143) and Ikerbasque. 

BL is supported by Global Dairy Platform.

JT received salary from Dairy Management Inc. 

These parties provided funding for the study: Arla Foods, Dairy Australia, Dairy Companies of New Zealand, Dairy Management Inc., Global Dairy Platform, Global Round Table for Sustainable Beef, McDonalds Corporation, and Meat and Livestock Australia  

The funders had a role in the study design by providing some of the general questions. "

6. Please remove your figures from within your manuscript file, leaving only the individual TIFF/EPS image files, uploaded separately. These will be automatically included in the reviewers’ PDF.

Additional Editor Comments:

1. Please revise the paper title and improve the readability of the paper title.

2. Please make sure that the format ot fhis paper is consistent with the format requirements of PLOSOne.

3. Please improve the quality of figures.

Reviewers' comments:

Reviewer's Responses to Questions

**Comments to the Author**

1. Is the manuscript technically sound, and do the data support the conclusions?

Reviewer #1: Yes

Reviewer #2: Yes

2. Has the statistical analysis been performed appropriately and rigorously? 

Reviewer #1: Yes

Reviewer #2: Yes

3. Have the authors made all data underlying the findings in their manuscript fully available?

Reviewer #1: Yes

Reviewer #2: Yes

4. Is the manuscript presented in an intelligible fashion and written in standard English?

Reviewer #1: Yes

Reviewer #2: Yes

5. Review Comments to the Author

Reviewer #1: Using a simple GWP*-TCRE framework together with a set of scenarios/assumptions, the authors presented the estimates for the past and projected contribution of livestock/ruminant CH4 emissions to the global temperature. The GWP*-TCRE framework is a simplified way to better, even not accurate, quantify the global warming potential for short-lived greenhouse gases like CH4. The topic and the results are interesting and useful for informing climate mitigation pathways involving all major GHGs. The manuscript deserve a publication after some revision. My a few concerns are as follows.

(1) Warming contribution from CH4 emissions from livestock sector is the main focus of this study. However, the definition of the CH4 emissions used is confusing across the text. “Livestock”, “cattle”, “cattle and buffalo” were all used casually in the current text, though the emission quantity can be quite different. Livestock emissions including ruminant and monogastric ones; cattle (or beef cattle and dairy cows) and buffaloes are ruminants, while sheep and goats are also ruminants that emitted a substantial part of the livestock CH4 emissions; CH4 emissions are also from pig and poultry, especially manure management. In addition, there are enteric CH4 emissions and those from manure management. It is absolutely necessary to clarify in every place of the manuscript what kinds of CH4 emissions (livestock, ruminant, or cattle only; enteric or including manure management) were accounted and used in the calculation of GWP* and their TCRE. It can also be clarified in a paragraph in the Materials and Methods, and used consistently across the manuscript.

(2) A lot of scenarios were set in this study to illustrate the effects of CH4 emissions. Readers will probably get lost during the long description between L197 and L261. It is necessary to provide an overview figure or table to illustrate the setups and especially the rational and the purpose. People can come back to the figure or table when read the super long results section.

(3) Most of the figures were consisted with a single line or with several subplot presenting similar information. It is recommended to combine some of the figures e.g., Fig. 1 and 3, or improve the quality and the presentation of the figures, e.g., subplots can be combined into one with three bars for each sensitivity period in Fig. 7. Most current figures were not publishable for their current form.

(4) The text is unnecessarily long, which reads like a boring report without focus. I would strongly suggest a substantial shortening of the Results and Discussion section, showing only the core findings and information. For example, numbers that do not trigger further discussion or implications could be neglected. Paragraph between L395 and 405 seems redundant (similar information has been presented in previous sections. Furthermore, there are quite some paragraph that cite several papers with too much details. The text should be precise and concise.

(5) A few important terms may be mis-used or mis-phrased. For example, it is not clear what does the “sustained” mean. Sustainable? Or constant? L389 “Tier” rather than “tear level”.

Reviewer #2: Your research is interesting, but there are still some problems that confuse me.

1.You mentioned that “These differences are hidden when describing the effects of climate mitigation using annual CO2-e emissions and when targets are based on aggregated annual emission rates” in the research, but I don’t understand “the effects” and the reasons for “the effects”.

2.You mentioned that “Sustained annual reductions in CH4 emissions of 0.31% by the global cattle sector would stabilize their future effect on global temperature while greater reductions would reverse historical past contributions to global warming by the sector in a similar fashion to increasing C sinks”. But I don't know why emitting less CH4 emissions (greater reductions of 0.31%) rather than negative CH4 emissions would “reverse historical temperature impacts from previous decades”. Can you provide some level of details about the conclusion?

3.The feasibility of the target (annual reductions in CH4 emissions of 0.31% by the global cattle sector) and the related measures need to be taken, such as a shift to sustainable diets.

4.The blue line (mentioned in line 559, pp. 34) does not appear in the figure 9.

6. PLOS authors have the option to publish the peer review history of their article (what does this mean?). If published, this will include your full peer review and any attached files.

Reviewer #1: No

Reviewer #2: No

---

## [Author Response · Author response to Decision Letter 0]

13 Jun 2023

COMMENTS TO THE AUTHOR

REVIEWER #1: 

Using a simple GWP*-TCRE framework together with a set of scenarios/assumptions, the authors presented the estimates for the past and projected contribution of livestock/ruminant CH4 emissions to the global temperature. The GWP*-TCRE framework is a simplified way to better, even not accurate, quantify the global warming potential for short-lived greenhouse gases like CH4. The topic and the results are interesting and useful for informing climate mitigation pathways involving all major GHGs. The manuscript deserve a publication after some revision. 

Thanks for this positive feedback.

My a few concerns are as follows:

(1) Warming contribution from CH4 emissions from livestock sector is the main focus of this study. However, the definition of the CH4 emissions used is confusing across the text. “Livestock”, “cattle”, “cattle and buffalo” were all used casually in the current text, though the emission quantity can be quite different. Livestock emissions including ruminant and monogastric ones; cattle (or beef cattle and dairy cows) and buffaloes are ruminants, while sheep and goats are also ruminants that emitted a substantial part of the livestock CH4 emissions; CH4 emissions are also from pig and poultry, especially manure management. In addition, there are enteric CH4 emissions and those from manure management. It is absolutely necessary to clarify in every place of the manuscript what kinds of CH4 emissions (livestock, ruminant, or cattle only; enteric or including manure management) were accounted and used in the calculation of GWP* and their TCRE. It can also be clarified in a paragraph in the Materials and Methods, and used consistently across the manuscript.

We agree that it may be confusing why we use different databases for the different analysis. We used livestock CH4 emissions data (all species, enteric and manure) for the long-term analysis (1750-2019) and global cattle (defined as dairy and beef cattle, and buffalo-also enteric and manure, according to FAOstat stats) for shorter analysis (1961-2019). Our main focus would have been “global cattle” if we had had long-term (1750-2019) stats but the long-term stats (1750-2019) are only available for aggregated livestock. Our analysis included both enteric CH4 and manure management CH4. 

We have tried to clary these points in the text: We have this text in the material and methods section:

L104-108. “the retrospective warming added by CH4 emissions from livestock in aggregate and from global cattle (as defined by dairy and beef cattle and buffalo) globally between 1750-2019 and 1961-2019, respectively, and (iii) analyse, using these metrics, the implications on future global temperature change of adopting different CH4 mitigation strategies that lead to warming stabilisation by the global cattle industry.”

L114-118 “It must be noted that we used total livestock (including all animal species) CH4 emissions for the long-term retrospective analysis (1750-2019) since data disaggregating livestock emissions by species (i.e. cattle) were not available for the years before 1961. For the short-term analysis (1961-2019) we used historical cattle (dairy and beef) and buffalo CH4 emissions (all together referred as “global cattle”).” 

We also explained that short term analysis data for CH4 referred to cattle (dairy and beef) and buffalo CH4 emissions together and used the term “Global Cattle” for simplicity purposes meaning both cattle and buffalo. 

L117-L118 “For the short-term analysis (1961-2019) we used historical cattle (dairy and beef) and buffalo CH4 emissions (all together referred as “global cattle”).” 

L124-L126 “Short-term (1961-2019) historical cattle and buffalo annual enteric and manure CH4 emissions were obtained from the FAOSTAT database [11]. For simplicity purposes, “global cattle” refers to cattle (dairy and beef) and buffalo in this study.” 

For both livestock CH4 (long-term analysis) and Global Cattle CH4 (Short-term analysis), we accounted for both of the main sources of CH4: enteric and that from manure management. This is mentioned in:

L119-120. “Long-term (1750-2019) historical livestock annual enteric and manure CH4 emissions data were obtained from…”: 

L124-L125. “Short-term (1961-2019) historical cattle and buffalo annual enteric and manure CH4 emissions were obtained from the FAOSTAT database [11]. “ 

L164-L166. In figure 2 caption we also now refer to “Global Cattle” for uniformity of terms: New “Fig 2. Methane emissions from global cattle (aggregating dairy and beef cattle and buffalo) (a)…”. 

(2) A lot of scenarios were set in this study to illustrate the effects of CH4 emissions. Readers will probably get lost during the long description between L197 and L261. It is necessary to provide an overview figure or table to illustrate the setups and especially the rational and the purpose. People can come back to the figure or table when read the super long results section.

Thanks for this recommendation. We have elaborated a new table (now Table 1) to clarify the characteristics of each of the scenarios analyses.

(3) Most of the figures were consisted with a single line or with several subplot presenting similar information. It is recommended to combine some of the figures e.g., Fig. 1 and 3, or improve the quality and the presentation of the figures, e.g., subplots can be combined into one with three bars for each sensitivity period in Fig. 7. Most current figures were not publishable for their current form.

First, we rearranged all figures for best format so now they all have good resolution and 600 dpi. As recommended, we combined Fig 1 and 3 into new Fig 1 and Fig 7 has been combined into one figure (now new Fig 5). 

(4) The text is unnecessarily long, which reads like a boring report without focus. I would strongly suggest a substantial shortening of the Results and Discussion section, showing only the core findings and information. For example, numbers that do not trigger further discussion or implications could be neglected. Paragraph between L395 and 405 seems redundant (similar information has been presented in previous sections. Furthermore, there are quite some paragraph that cite several papers with too much details. The text should be precise and concise.

The text has been shortened as suggested and some citations have been taken away. We also changed the level one headings to match the subtitles within this section to the 5 analysis scenarios listed in the M&M section to make it easier for readers to follow.

Some IPCC-based text from the Paragraph L395-405 could be considered as justification for this analysis, so we moved it from the discussion point to the introduction (L81-L90) : “It is recognized that GWP100 shows…”.

(5) A few important terms may be mis-used or mis-phrased. For example, it is not clear what does the “sustained” mean. Sustainable? Or constant? L389 “Tier” rather than “tear level”.

“Sustained” was meant to be used for reductions in emissions that are constant over time. Now we have amended and clarify this in all text. For all scenarios we have explicitly indicated the % of increase/decrease in annual emissions. For instance, for case study 5:

L269-L281.

“Analyzing the projected global warming associated with 3 futures global CH4 cattle emission pathways leading to a stabilisation of global temperatures at year 2050 (1990-2050 period) varying in the time and intensity when absolute CH4 emission rates are reduced as follows: 

(i) fast: large reductions (greater than those required for sustained emission rate reductions: 0.5% decrease per year) in global CH4 cattle emissions are introduced during the first decades (2020-2040) followed by reductions at smaller rates (2040-2050) (0.1% decrease per year).

(ii) sustained: reductions in global CH4 cattle emissions in a sustained way starting in 2020 (0.32% decrease per year, constant over time) 

 (iii) delayed: increase in absolute emission rates for the first 2 decades (2020-2040) (0.25% increase per year), followed by large reductions (0.84% decrease per year) in global CH4 cattle emissions are introduced during the last decade (2040-2050). “ 

“Tear” was a typo. Now we amended to Tier: “depending on the level of methodological complexity used (also denoted as tier level by IPCC (2019)”

REVIEWER #2: 

Your research is interesting, but there are still some problems that confuse me.

Thanks for this feedback, we will try to clarify the text to deal successfully with your confusion.

1.You mentioned that “These differences are hidden when describing the effects of climate mitigation using annual CO2-e emissions and when targets are based on aggregated annual emission rates” in the research, but I don’t understand “the effects” and the reasons for “the effects”.

L76-78. The sentence has been changed for clarification to. “These differences in temperature change are hidden when calculating and using annual CO2-e emissions to describe the impact of mitigation and targets based on aggregated annual emission rates”

2.You mentioned that “Sustained annual reductions in CH4 emissions of 0.31% by the global cattle sector would stabilize their future effect on global temperature while greater reductions would reverse historical past contributions to global warming by the sector in a similar fashion to increasing C sinks”. But I don't know why emitting less CH4 emissions (greater reductions of 0.31%) rather than negative CH4 emissions would “reverse historical temperature impacts from previous decades”. Can you provide some level of details about the conclusion?

First, we updated the % to 0.32% as 0.31% had been obtained with a previous formulation of the GWP* metrics and used in our first manuscript submissions by mistake (only for this case study, the rest of the study GWP* had been used consistently with the latest GWP* version of the formulation). 

A reduction in annual emission rates will have a different effect on the level of GHG (concentration) in the atmosphere depending on whether a GHG is a short-lived gas (e.g. CH4) or a long-lived gas (e.g. CO2). If it is a long-lived gas, the CO2 level (or concentration) and hence the amount of associated warming to this CO2 concentration in the atmosphere will get to a stable condition (for the first years at least, before starting to decline very slowly) only if we reduce CO2 emissions (net) to 0. So we reach an equilibrium between emissions and sinks. This is what is referred as net-0 CO2. For short-lived gases (e.g. CH4), CH4 is mostly removed from the atmosphere by chemical reaction (reacts with OH- for example) much faster than CO2, persisting for about 12 years (instead of 100s or even 1000s of years as CO2). Hence, small reductions in annual CH4 rates will already lead to a stabilisation of the CH4 concentration in the atmosphere, leading to heat trapping capacities that could have been found decades before that time (reversing historical temperature impacts). This is briefly mentioned in the Introduction section: 

L 71-L76 “Whereas methane (CH4)’s impacts on temperature varies strongly with time after emissions occur due to its short atmospheric life, CO2’s impact on temperature remains relatively constant for hundreds of years after the emission occurs [7]. Also, CO2, once emitted, leads to increasing global temperature until net-zero CO2 emissions are reached. By contrast, reductions in CH4 emissions lead to reversing warming within a few decades.” 

This is very well explained in [7] (Cain et al. 2019. npj Clim Atmos Sci. 2019;2: 1–7) in relation with how GWP* metrics can help mimic these GHG different dynamics for short and long-lived GHG. The concept is well known as indicated in Cain et al 2019 and expressed in the text. 

L78-80. “In fact, GWP* was not developed to capture this behaviour, which was already well known, but to demonstrate that it could be quantified relatively simply while continuing to use the already familiar GWP concept” 

This is also well explained in recent articles like Cain et al et al. 2021 [6] Paragraph from this article (but not in our article): “CO2 emissions continue increasing global mean temperature until net-zero emissions are reached, with potential for lowering temperatures with net-negative emissions. By contrast, reducing CH4 emissions starts to reverse CH4-induced warming within a few decades. 

Coming from this Cain et al. 2021 [6] we added this paragraph in the M&M section:

L196-L199. Emissions are used from years t (the year for which CO2-we emissions are being calculated) and from t-20 (20 years prior). This allows GWP* to represent the impact of new emissions (which cause strong additional warming), stable emissions (minimal additional warming) and reducing emissions (which reverses some past warming) [6].

The concept is very much shown as results and discussion in new Fig 7, where whereas reduction in CO2 emissions by 1% would still lead to temperatures until 2100 (from about 0.7oC in 2020 to 1.62 oC respect pre-industrial times). The same reduction (1%) for CH4 rates would lead to temperatures from 0.15 oC in 2020 to 0.12 oC in 2020, thus meaning a certain temperature impact reverse to the level of warming that CH4 had caused until the 1970s. 

3.The feasibility of the target (annual reductions in CH4 emissions of 0.31% by the global cattle sector) and the related measures need to be taken, such as a shift to sustainable diets.

Even though assessing the feasibility of GHG reductions per se is beyond the scope of this article, there is some reference to the type of strategies required and where these strategies may or may not apply: 

L502-l513. “Reisinger et al. [19] advocated for strategies to reduce CH4 emissions through more efficient production, technological advances and demand side changes, and importantly always carefully considering their interactions with land-based C sequestration. The challenge is that projected growth in livestock production systems and current CH4 emissions from ruminant livestock are greatest in low-and middle-income countries where the nutrient supply to the population could be severely compromised if animal-derived food demand is dramatically shortened. In addition, most production-side mitigation strategies that were designed for developed countries don’t apply to the production systems used in low- and middle-income countries [31] [32]. These socioeconomic and environmental aspects of mitigation were ignored in some studies [33] that explored the climate impact of implementing even more drastic scenarios, such as eliminating animal agriculture.”

Additionally, we added a reference in L589 pointing at to some recent study [37] that has evaluated (towards the 1.5-2 oC temperatures objectives of the Paris Agreement) the scope of adopting mitigation measures to reduce CH4 emissions from livestock.

4.The blue line (mentioned in line 559, pp. 34) does not appear in the figure 9.

There was a mistake in the final stage of the figure formatting. Now the figure (new Fig 8b) shows a blue line.

---

## [Decision Letter · Decision Letter 1]

26 Jun 2023

Retrospective and projected warming-equivalent emissions from global livestock and cattle calculated with an alternative climate metric denoted GWP*

PONE-D-23-08709R1

Dear Dr. del Prado,

We’re pleased to inform you that your manuscript has been judged scientifically suitable for publication and will be formally accepted for publication once it meets all outstanding technical requirements.

Kind regards,

Zhaoxia Guo

Academic Editor

PLOS ONE

Additional Editor Comments (optional):

Reviewers' comments:

Reviewer's Responses to Questions

**Comments to the Author**

1. If the authors have adequately addressed your comments raised in a previous round of review and you feel that this manuscript is now acceptable for publication, you may indicate that here to bypass the “Comments to the Author” section, enter your conflict of interest statement in the “Confidential to Editor” section, and submit your "Accept" recommendation.

Reviewer #1: All comments have been addressed

Reviewer #2: All comments have been addressed

2. Is the manuscript technically sound, and do the data support the conclusions?

Reviewer #1: Yes

Reviewer #2: Yes

3. Has the statistical analysis been performed appropriately and rigorously? 

Reviewer #1: N/A

Reviewer #2: Yes

4. Have the authors made all data underlying the findings in their manuscript fully available?

Reviewer #1: Yes

Reviewer #2: Yes

5. Is the manuscript presented in an intelligible fashion and written in standard English?

Reviewer #1: Yes

Reviewer #2: Yes

6. Review Comments to the Author

Reviewer #1: The authors well addressed my concerns. I do not have further concerns. This manuscript is a nice piece that worth to be published.

Reviewer #2: (No Response)

7. PLOS authors have the option to publish the peer review history of their article (what does this mean?). If published, this will include your full peer review and any attached files.

Reviewer #1: No

Reviewer #2: No

---

## [Editor Report · Acceptance letter]

14 Jul 2023

PONE-D-23-08709R1 

Retrospective and projected warming-equivalent emissions from global livestock and cattle calculated with an alternative climate metric denoted GWP* 

Dear Dr. del Prado:

I'm pleased to inform you that your manuscript has been deemed suitable for publication in PLOS ONE. Congratulations! Your manuscript is now with our production department. 

Kind regards, 

on behalf of

Dr. Zhaoxia Guo 

Academic Editor

PLOS ONE